# High Tuberculosis Preventive Treatment Uptake and Completion Rates Using a Person-Centered Approach among Tuberculosis Household Contact in Yogyakarta

**DOI:** 10.3390/tropicalmed8120520

**Published:** 2023-12-14

**Authors:** Felisia Felisia, Rina Triasih, Betty Weri Yolanda Nababan, Guardian Yoki Sanjaya, Setyogati Candra Dewi, Endang Sri Rahayu, Lana Unwanah, Philipp du Cros, Geoffrey Chan

**Affiliations:** 1Centre for Tropical Medicine, Faculty of Medicine, Public Health, and Nursing, Gadjah Mada University, Sleman 55281, Yogyakarta, Indonesia; ztb.fkkmk@ugm.ac.id; 2Department of Pediatric, Faculty of Medicine, Public Health and Nursing, Gadjah Mada University, Sleman 55281, Yogyakarta, Indonesia; 3Department of Health Policy and Management, Faculty of Medicine, Public Health and Nursing, Gadjah Mada University, Sleman 55281, Yogyakarta, Indonesia; gysanjaya@ugm.ac.id; 4Yogyakarta City Health Office, Yogyakarta 55165, Yogyakarta, Indonesia; setyogati.candra@jogjakota.go.id (S.C.D.); endangsr@jogjakota.go.id (E.S.R.); lana@jogjakota.go.id (L.U.); 5Tuberculosis Elimination and Implementation Science Working Group, Burnet Institute, Melbourne, VIC 3004, Australiageoff.chan@burnet.edu.au (G.C.)

**Keywords:** tuberculosis preventive treatment (TPT), short regimen, person-centered care

## Abstract

Coverage of tuberculosis preventive treatment (TPT) in Indonesia is inadequate, and persons who start TPT often do not complete treatment. In 2020, Zero TB Yogyakarta implemented person-centered contact investigation and shorter TPT regimen provision in collaboration with primary health care centers. Between 1 January 2020 and 31 August 2022, we assessed eligibility for TPT among household contacts of persons with bacteriologically confirmed TB (index cases) and offered them a 3-month TPT regimen (3RH or 3HP). A dedicated nurse monitored contacts on TPT for treatment adherence and side effects every week in the first month and every two weeks in the next months. Contacts were also able to contact a nurse by phone or ask for home visits at any point if they had any concerns. A total of 1016 contacts were eligible for TPT: 772 (78.8%) started short regimen TPT with 706 (91.5%) completing their TPT. Side effects were reported in 26 (39%) of the non-completion group. We conclude that high rates of TPT uptake and completion among contacts assessed as eligible for TPT can be achieved through person-centered care and the use of shorter regimens. Side-effect monitoring and management while on TPT is vital for improving TPT completion.

## 1. Introduction

As the third largest contributor to the global TB burden, Indonesia has a TB incidence of 354 patients per 100,000 people [1]. In high-burden settings, household contacts of TB cases are at higher risk of infection [2,3,4,5]. Individuals with latent TB infection (LTBI) are infected but do not currently have active disease and have a 5–10% lifetime risk of progressing from LTBI to active disease, with the majority of such progression occurring within the first two years after infection [6].

TB preventive treatment (TPT) uses medication to treat infection in people with LTBI. The End TB Strategy highlights the importance of TPT in one of the pillars of its TB elimination strategy. Moreover, commitment to accelerate efforts to end TB was addressed in the United Nations High Level Meeting on TB of 2018 and 2023. One of the targets is to improve TPT coverage to at least 90% by 2027. Modeling studies estimate that the optimal implementation of TPT in identified high-risk groups—such as people living with HIV (PLHIV) and household contacts (HHCs)—has the potential to reduce the annual TB incidence by 8.3% relative to 2015 [7]. Moreover, cost-effectiveness modeling has shown that the investigation of contacts with the provision of short-course preventive treatment for TB compares favorably with other interventions in terms of value [8]. Despite this, the coverage of TPT globally is inadequate, and in Indonesia, only 0.2% of HHCs were given TPT in 2020 and 2021 [9,10,11].

A further challenge is that persons who start TPT often do not complete treatment. A 2016 meta-analysis reported only 52% TPT completion in low–middle-income countries [12]. Shorter regimens with a 3–4-month duration have been reported to offer higher completion than longer regimens [13,14]. In Indonesia, TPT completion was 78.7% using a four months of daily rifampicin (4R) regimen in Bandung, and 25.6% using six months of daily isoniazid (6H) [15,16]. In 2020, the Indonesian National TB Program (NTP) recommended TPT for HHCs of all ages, using shorter TPT regimens of three months of daily isoniazid and rifampicin (3RH) or three months of weekly isoniazid and rifapentine (3HP). This replaced its previous recommendations for the use of 6H only in close contacts under five years of age. To date, there are limited data on the use of these regimens in Indonesia.

Common barriers documented in providing TPT include poor patient acceptance, poor treatment adherence, and access difficulties [8,9]. Overcoming these barriers is needed to achieve better outcomes with TPT. Alongside the choice of regimen, effective person-centered care for TPT provision is an important consideration in supporting TPT completion. This involves tailoring the treatment to the individual needs, preferences, and circumstances of the persons receiving TPT. In practice, this entails the assessment of individual’s risk factors and medical history; encouraging shared decision making between healthcare providers, individuals, and their support networks; the consideration of cultural, social, and economic context; flexible service delivery; and maintaining good communication by addressing individual’s concern and questions promptly throughout the treatment process [17,18,19]. Implementing this approach has the potential to positively impact treatment outcomes.

In 2020, the Zero TB Yogyakarta (ZTBY) project commenced an implementation study to accelerate TB elimination in Yogyakarta. This project conducted community-based active case finding (ACF) and implemented TB contact investigation and TPT provision in collaboration with primary health care (PHC) centers. In this study, we evaluated the treatment outcomes from the Zero TB Yogyakarta model of person-centered TPT provision for HHCs using short regimens.

## 2. Methods

### 2.1. Study Design and Population

This was a cross-sectional study using routinely collected secondary data of HHCs living in Kulonprogo and Yogyakarta investigated by ZTBY contact investigation teams. The study population consisted of persons who were investigated by ZTBY contact investigation services between 1 January 2020 and 31 August 2022 due to them being household contacts of a bacteriologically confirmed TB index case (although some participants in this study then initiated TPT after 31 August 2022). The date of notification in the national TB data system for the index case from whom these contacts were identified ranged between 1 January 2018 and 31 August 2022.

### 2.2. General Setting

The Special Region of Yogyakarta is a province located in south-central Java, Indonesia with a population of 4 million. The province is divided into five regions, ranging from urban to rural, as illustrated in Figure 1 [20]. The investigation of HHCs and provision of TPT with short regimens is recommended in NTP guidelines. However, prior to 2020, contact investigation and TPT provision were not routinely implemented in Yogyakarta. In 2020, only 0.56% of HHCs were given TPT in Yogyakarta Province. Moreover, the NTP only made short TPT regimens routinely available through government health services in March 2022 in Yogyakarta Province. Hence, the provision of shorter TPT regimens described in this study preceded their roll-out as standard care through the NTP.

### 2.3. Specific Setting

Zero TB Yogyakarta implemented the study in two regions in Yogyakarta Province: Yogyakarta (urban) and Kulonprogo (rural). The project implemented contact investigation and TPT provision in these districts commencing in 2020. Index cases notified between 1 January 2018 and 31 August 2022 and residing in these regions were identified from health facility TB treatment registers. Operationally, ZTBY contact investigation was conducted for household contacts of bacteriologically and clinically diagnosed index cases. However, only contacts of bacteriologically confirmed cases are included in this study.

Between 1 January 2020 and 31 August 2022, two contact investigation teams, each consisting of one ZTBY nurse, one or more PHC center staff and/or community health workers visited the household. Eligible index cases who consented to investigation were asked to list HHCs. HHCs were defined as persons living in the index case’s household in the three months prior to their TB diagnosis. HHCs were screened if present. Repeat visits and phone consultation were used to investigate contacts not present at the initial household visit. For HHCs no longer living in the target districts, the contact investigation team reported the names and addresses (if known) to the District Health Office.

HHCs were screened for TB symptoms and tested for TB infection using a tuberculin skin test (TST) or interferon-gamma release assay (IGRA), if indicated. During the study, we piloted the use of the IGRA for a period of 6 months. Contacts who were investigated during this period were tested using IGRA if they were over the age of 6 years, claimed to not have HIV or diabetes mellitus, and provided consent. Contacts who were investigated outside the IGRA pilot period, did not meet the criteria for IGRA testing, or refused IGRA procedure were tested using TST. TB infection testing was indicated for contacts above five years old with bacteriologically confirmed index cases and for contacts under five years old with clinically confirmed index cases. All HHCs were then referred for CXR to ACF sites or to the nearest CXR facility, irrespective of the screening and TB infection testing results. HHCs who screened positive based on symptoms or CXR result had a spot sputum sample collected for GeneXpert MTB/RIF testing (Cepheid, Sunnyvale, CA, USA).

Final diagnosis and eligibility for TPT were determined by health facility doctors, in collaboration with project doctors, internists, and pediatricians. HHCs aged five years and older were eligible for TPT if active TB was ruled out and their TB infection testing result was positive. Children younger than five years were eligible for TPT if active TB disease was ruled out and if the index case was bacteriologically confirmed. TPT eligibility in children under five with a clinically confirmed index case required a positive TB infection testing result with no evidence of active disease.

Contacts eligible for TPT were referred to health centers for further history taking and physical examination. TPT was not commenced if HHCs had contraindications, had already completed TPT within the last 5 years with no new TB contact, or if the clinical review for TPT initiation found that the risk outweighed potential benefit. In this clinical review, a PHC doctor assessed the eligible contact and determined whether the potential risks associated with administering TPT were greater than the expected benefits for each individual. This included, for example, possible side effects of the medication, possible interactions with other medications, or health conditions that may exacerbate due to TPT administration.

The first dose of TPT was observed directly, and was then followed by self-administered therapy. A follow-up visit by a ZTB nurse was scheduled in the first week after a contact started TPT. First-week visits included screening for side effects and evaluation of treatment adherence by pill count. Contacts were also asked if they had any concerns regarding the treatment. If the treatment was going well, contacts were scheduled to be monitored every week in the first month, then every two weeks in subsequent months until treatment completion. Follow-up was performed either by phone or in-person visit, depending on the contact’s preference. Contacts were also able to call the ZTB nurse by phone or ask for home visits at any point if they had side effects or concerns. The management of minor side effects and counseling were provided by the ZTB nurse. For moderate to severe side effects, or if there were further concerns regarding TPT, contacts were referred to their nearest health facility.

Treatment outcome was categorized into complete and not complete. Contacts were directed to take all the prescribed doses, 84 doses for 3RH and 12 doses for 3HP, and completion of TPT was defined as at least 80% intake of 3RH course or 11 doses of 3HP within 120 days of starting treatment. Reasons for non-completion were recorded as treatment failed, resulted in death, and treatment discontinued. In instances of treatment discontinuation, the reasons for stopping treatment were recorded as follows: the presence of drug side effects with clinician’s decision to discontinue the treatment; loss to follow-up; patient’s decision to discontinue the treatment; and other reasons. Where there was no outcome recorded for a patient, the outcome was considered as not evaluated and categorized as not complete.

### 2.4. Data Variables, Sources of Data, and Data Collection

The ZTBY project records patient care data for contact investigation and TPT data using REDCap electronic data capture tools hosted by the Universitas Gadjah Mada. Study data were extracted from REDCap on February 2023. This included demographic information (age, gender, nutritional status, and district address); LTBI test result; and TPT eligibility, initiation, and follow-up details.

### 2.5. Analysis and Statistics

Stata Version 17 (StataCorp, College Station, Texas) and R version 4.2.2 (R Foundation for Statistical Computing, Vienna) were used to clean and analyze the data. Frequency and proportions were prepared for descriptive analysis. Univariate and multivariate logistic regression were used to analyze risk factors for TPT non-completion.

## 3. Results

As illustrated in Figure 2, our study included 1016 household contacts assessed as eligible for TPT. The eligible contacts in our study were identified from the screening of 2813 household contacts of 915 bacteriologically confirmed index cases. Of the 1016 contacts eligible for TPT, 801 (78.8%) of them started TPT. Of those who started TPT, 772 (96.4%) were started on short regimens. There were 29 contacts started on TPT who were excluded from the study because they were on other regimens—7 on 6H; 21 on 6LFX (6 months of levofloxacin ± ethambutol); and 1 on 4R. The characteristics of contacts starting short TPT regimens are shown in Table 1. In our study, 3RH was given to 615 (76.8%) contacts and 3HP to 157 contacts (19.6%). Overall, 706 (91.5%) contacts on short regimens completed TPT, while 66 (8.5%) did not complete TPT. For the 790 contacts for whom both the exact date of the index case notification and date of TPT initiation were recorded, the median time from index case notification to TPT initiation was 204.0 days (IQR 74.0–576.0).

Risk factors for non-completion of short TPT regimens are shown in Table 2. Completion rates were lower for 3HP than for 3RH but were not found to be significant in multivariate analysis. In the univariate analysis of risk factors for treatment completion amongst those prescribed TPT, the reporting of any side effects was the only measured variable associated with non-completion (odds ratio 1.89 confidence interval (CI) 1.11–3.16, *p*-value 0.017). This association persisted in multivariate analysis (adjusted odds ratio 3.59 CI 1.80–7.29, *p*-value < 0.001).

Among the 66 contacts with an outcome classified as ‘not complete’, 19 contacts did not undergo evaluation. For the remaining 47 contacts, 1 died, 1 exhibited treatment failure, and 45 discontinued treatment. Among those who discontinued treatment, 22 (48.9%) were stopped due to a clinician’s decision in response to managing side effects, 4 (8.9%) were lost to follow-up, 17 (38%) stopped of their own accord, and an additional 2 (4.4%) discontinued for other reasons, i.e., stopped due to midwives’ decision during pregnancy and a change of regimen following the discovery that the index case had multidrug-resistant tuberculosis (MDR-TB).

For each type of side effect that was reported, Table 3 shows the proportion of contacts on each regimen who ever reported the side effect while on preventive treatment. Among those who reported side effects, Figure 3 shows the month in which the contact first reported this side effect for each side effect. Among most of the contacts reporting any of the recorded side effects, their first report of the side-effect was in Month 0, which was the first month on TPT.

## 4. Discussion

This study is among the first to report on the operational use of shorter three-month regimens for TB preventive treatment in Indonesia. There was a high level of uptake of TPT (80%) among the contacts eligible for TPT from Zero TB contact investigation. The observed uptake of TPT reflects the acceptance of TPT among those who completed assessment and were assessed as eligible for TPT, rather than among all contacts who were identified and investigated. Reducing the loss to care in earlier stages of the cascade would be needed to accurately assess eligibility among all contacts and acceptability of TPT as it might differ among those who may have been eligible but did not complete assessment. Nonetheless, this reflects the good acceptability of TPT with shorter regimens among contacts who do complete assessment. Of note, this level of acceptance was achieved despite the median of 204 days from the time of index case notification to TPT initiation in our study. This delay occurred because for the contact investigation we conducted from 2020 to 2022, we included contacts of index cases notified as far back as 2018. This was a deliberate inclusion so as to catch up on contacts who were missed prior to the introduction of contact investigation services.

Completion of TPT among contacts on short regimens was over 90%. These completion rates for short regimens compare favorably to those observed under trial study conditions in Indonesia and elsewhere, and to results with longer regimens reported in Indonesia, i.e., 78.7% using 4-month daily rifampicin (4R) and 65.5% using 9-month daily isoniazid (9H) [15,16,21,22]. The proportions of uptake and completion in our study were higher than those found with 3HP implementation in two urban centers in Pakistan, where 59% initiated and 69% completed 3HP [23]. These findings highlight the potential of the Zero TB person-centered model using shorter TPT regimens to achieve good coverage and outcomes, based on data from over two years of TPT provision to household contacts. Importantly, the results reflect “real-world” program conditions with contacts in urban and rural settings.

In comparison to 3RH, 3HP had lower completion rates, but this difference was not found to be significant in multivariate analysis. However, missing outcome data (categorized as “not evaluated” in our analysis) made up a higher proportion of the unfavorable outcomes for 3HP. This may reflect a delay in obtaining outcome data for more recent contacts, the majority of whom were on 3HP, rather than indicating that completion is worse among contacts on 3HP. While both regimens are recommended in Indonesian guidelines, there is a preference for 3HP as the weekly dosing is thought to be more convenient. In this study, both regimens had high completion rates with self-administered treatment with regular telephone support and good education.

The reporting of a side effect was significantly associated with the non-completion of TPT (AOR 3.59, CI 1.80–7.29, *p*-value < 0.0001). This is consistent with the recorded reasons for stopping treatment, with clinicians deciding to stop TPT due to side effects in 49% of the contacts who did not complete TPT. For most side effects, there was a slightly higher proportion of contacts on 3RH reporting the side effect than for contacts on 3HP. There was one death recorded on TPT, but this was deemed as not related to TB or TB preventive treatment. The one contact diagnosed with TB while on TPT was diagnosed within two weeks of starting TPT which likely reflects inaccurate rule out of active disease prior to starting TPT.

Importantly, 27% (*n* = 207) of contacts on short regimens reported a side effect, and given the association with non-completion, improving support for contacts with side effects should be a priority. In the Zero TB model, there is a nurse who routinely monitors contacts for treatment uptake and side effects and provides psychological counseling and support at the same time. Adaptations to this existing role would be a logical entry point into further improving the management of contacts experiencing side effects.

A related finding is that among contacts reporting side effects, the first report of the side effect is usually in the first month on TPT. Our data collection on TPT side effects precludes the possibility of analyzing whether repeated reports of the same side effect constitute a single or multiple episodes. Nonetheless, our findings show that many contacts on short regimens first experience side effects within the first month on TPT. This might be caused either by the physical adaptation process that happens early in the treatment course, or the management of drug side effects that had succeeded in the first month. However, it could also reflect a decreased tendency to check for or report the side effect if it is first experienced in the second and third month of treatment, especially since scheduled monitoring interactions are less frequent after the first month on TPT. In order to achieve successful TPT scale up in Indonesia, clear protocols on assessing and managing side effects at different points during TPT are needed.

This is a retrospective cross-sectional study using data collected operationally. Missing data and inconsistent data were noted in the operational data relating to TPT eligibility. For follow up interactions of contacts on TPT, phone and home visits were recorded in the same way, and ad hoc contact-initiated phone contacts were recorded as well as scheduled follow-ups. Hence, the nature of monitoring and care could be quite different between contacts who have similar numbers of follow-up interactions.

The Zero TB Yogyakarta Project’s activities have relied on funding from external donors, and the replication of its model of care is likely to be challenging if reliant on recurrent health-service funding. The understanding of which components of the model are critical to its success will be an important first step in using the model to guide decisions on how to deliver TPT services in Indonesia. In our model of care, we assigned one nurse to communicate and follow up on contacts on TPT. This nurse provided education and support to contacts on TPT, with support being the most intensive in the first month that a person is on TPT. Each ZTBY nurse was able to support the HHCs from 4-7 primary health care centers. This model of TPT support, while additional to current resourcing within government health services, should be considered as a means of providing patient-centered TPT care in similar settings in Indonesia. Additionally, attention to local decision making and health system strengthening has been shown to improve uptake and completion rates during TPT scale up [24].

We recommend strengthening program monitoring, including routine gap analysis and the cascade of care during TPT scale up. While the TPT outcomes we report in this study compare well to global benchmarks and the End TB targets, there is still ample scope to improve the model of care to better support contacts on TPT. The evaluation of the project implementation could give information on different barrier-specific solutions in order to overcome the policy-practice gap and provide a functional TPT program for household contacts in endemic settings.

## 5. Conclusions

This study demonstrates that high rates of TPT uptake and completion can be achieved among eligible household contacts through person-centered care and the use of shorter regimens. Both 3HR and 3HP had similarly high rates of uptake and completion with self-administered treatment with regular supported care. Adapting the approach for contact monitoring and management while on TPT with a specific focus on management of side effects could better support contacts on TPT and help to further improve rates of completion.

## Figures and Tables

**Figure 1 tropicalmed-08-00520-f001:**
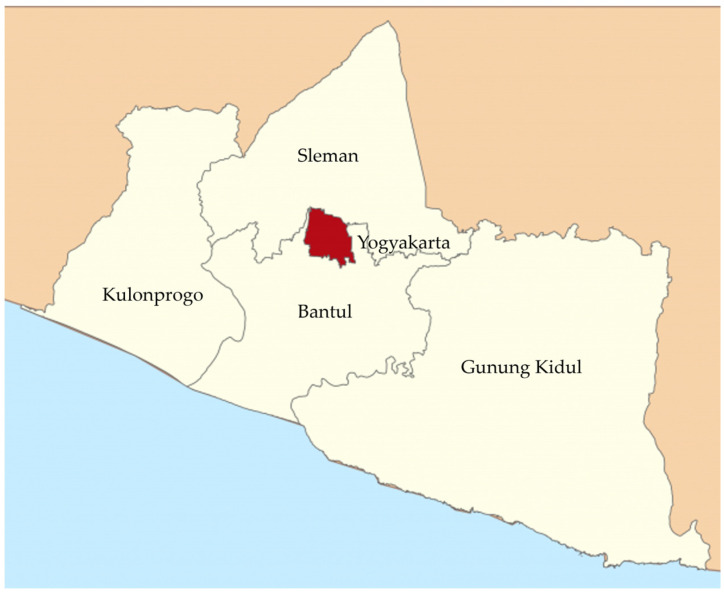
Map of Yogyakarta Province.

**Figure 2 tropicalmed-08-00520-f002:**
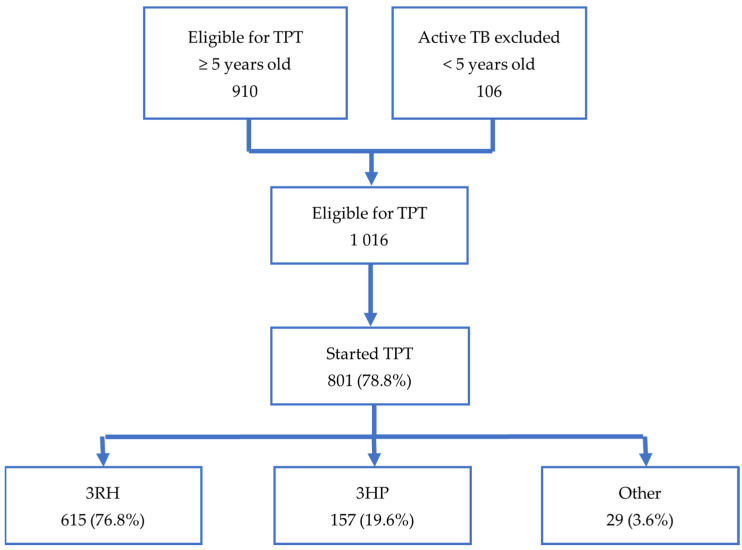
The cascade of care for TPT in YY and KP from January 2020 to August 2022.

**Figure 3 tropicalmed-08-00520-f003:**
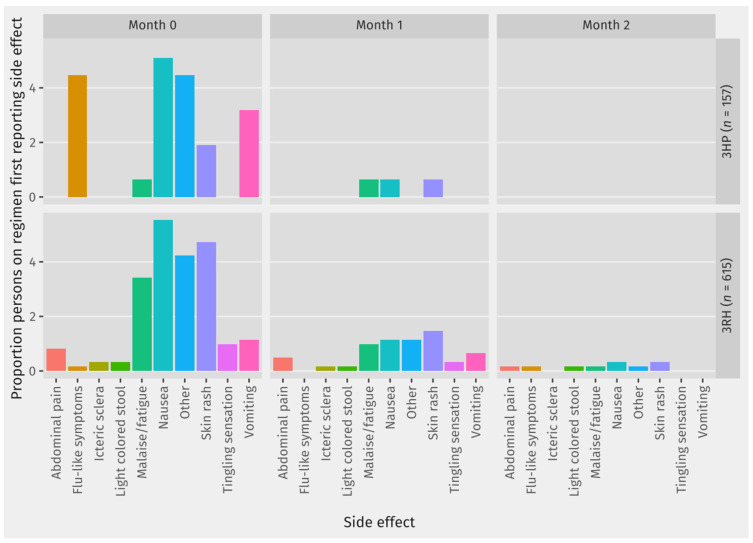
Proportion of contacts on short TPT regimens who reported side effects by month of first report of side effect.

**Table 1 tropicalmed-08-00520-t001:** Characteristics of contacts started on short TPT regimens.

Characteristic		Total *(n* = 772)
Age Group	<5 years	77 (10.0%)
	≥5 to 14 years	82 (10.6%)
	≥15 to 59 years	503 (65.2%)
	≥60 years	110 (14.2%)
Sex	Male	363 (47.0%)
	Female	409 (53.0%)
BMI	< 18.5	176 (24.2%)
	≥18.5 to 22.9	252 (34.6%)
	≥23 to 24	89 (12.2%)
	≥25	211 (29.0%)
District	Yogyakarta	494 (64.0%)
	Kulonprogo	278 (36.0%)

**Table 2 tropicalmed-08-00520-t002:** Risk factors for non-completion of short TPT regimens.

Summary Statistics				Univariate			Multivariate		
Characteristic		Complete *(n* = 706) ^1^	Not Complete (*n* = 66) ^1^	OR ^2^	95% CI ^2^	*p*-Value	AOR ^2^	95% CI ^2^	*p*-Value
Age (median, IQR)		39 (17, 54)	39 (20, 53)	1.00	0.99, 1.01	0.7	1.00	0.99, 1.02	>0.9
Sex	Male	338 (48%)	25 (38%)	—	—		—	—	
	Female	368 (52%)	41 (62%)	1.51	0.90, 2.56	0.12	1.19	0.63, 2.25	0.6
Regimen	3RH	568 (80%)	47 (71%)	—	—		—	—	
	3HP	138 (20%)	19 (29%)	1.66	0.93, 2.88	0.077	1.52	0.64, 3.34	0.3
District	Yogyakarta	444 (63%)	50 (76%)	—	—		—	—	
	Kulonprogo	262 (37%)	16 (24%)	0.54	0.29, 0.95	0.040	0.65	0.30, 1.3	0.3
Reported any side effect on TPT	No	525 (74%)	40 (61%)	—	—		—	—	
	Yes	181 (26%)	26 (39%)	1.89	1.11, 3.16	0.017	3.59	1.80, 7.29	<0.001
Scheduled vs. actual follow-up interactions *		−2.0 (−5.0, −1.0)	0.0 (−2.2, 0.0)	1.12	1.03, 1.23	0.010	1.03	0.94, 1.15	−0.5

^1^ Median (IQR), *n* (%). ^2^ OR = Odds ratio, CI = Confidence interval. * A negative number indicates less visits were conducted than were scheduled.

**Table 3 tropicalmed-08-00520-t003:** Number of persons reporting side effect at any time while on TPT.

Side Effect	3RH (*n* = 615) ^1^	3HP (*n* = 157) ^1^
Abdominal pain	9 (1.46%)	0 (0.0%)
Flu-like symptoms	2 (0.33%)	7 (4.46%)
Icteric sclera	3 (0.49%)	0 (0.0%)
Light-colored stool	4 (0.65%)	0 (0.0%)
Malaise/fatigue	28 (4.55%)	2 (1.27%)
Nausea	43 (6.99%)	9 (5.73%)
Skin rash	40 (6.5%)	4 (2.55%)
Tingling sensation	8 (1.3%)	0 (0.0%)
Vomiting	11 (1.79%)	5 (3.18%)
Other	34 (5.53%)	7 (4.46%)

^1^ *n* reporting side effect (proportion of persons on regimen).

## Data Availability

Due to data privacy concerns, data are not made publicly available. However, reasonable data requests may be granted through contacting the corresponding author.

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
