# Peer review of "High Tuberculosis Preventive Treatment Uptake and Completion Rates Using a Person-Centered Approach among Tuberculosis Household Contact in Yogyakarta"

_tropicalmed, 2023, doi:10.3390/tropicalmed8120520_

Round 1

Reviewer 1 Report

Comments and Suggestions for Authors

I really appreciate the authors for great work with this manuscript. This a relevant topic for TB control and has important programmatic implications for NTPs from LMICs. I have listed some observations below

Abstract: The abstract reads well. Some minor comments

1. Line 20: Use 'index patients' instead of index cases. The case is considered as a demeaning word and better to use patient (https://stoptb.org/assets/documents/resources/publications/acsm/LanguageGuide_ForWeb20131110.pdf). This comment is applicable to the whole manuscript. 

2. Line 20: Whether your study population is only bacteriologically confirmed TB patients? Later in the methods, you also talk about clinically diagnosed patients.

3. Line 26: Please provide what percentage had any side effect. 

4. Line 26 and 27: We need to be very careful before commenting on the uptake in this manuscript. There is large scope for selection bias, the authors are trying hard to highlight high uptake of TPT among this cohort and trying to attribute it to person-centered care. I think, limiting to conclusion on high completion rate and importance of adverse event monitoring should be sufficient. 

Introduction: The introduction reads and flows well. Some minor comments,

1. You can quote UNLHM of 2018 and 2023 to highlight the importance of TPT and improving the TPT coverage. Also, mention about suboptimal achievement of targets

2. Line 51: Please provide reference from global studies on better treatment completion rates with shorter TPT regimen.

3. Line 58: It is better to briefly describe the person-centered care for TPT provision. Please highlight how this model is able to overcome some of the barriers for optimal implementation of TPT for household contacts

4. Line 65-66: Your study objective is pretty clear. According to objectives, the focus is to estimate the treatment outcomes. I think restricting to objectives and may be assessing the factors associated with the unsuccessful treatment outcomes should be sufficient. However, authors have tried to comment about the TPT uptake without taking into account the attrition in the earlier stages of contact tracing and contact investigation. 

Methods: The methods section of the manuscript needs major revision.

1. Please bring the 'study design' subheading to the top.

2. Line 128: You mention this as a retrospective study, but it is not. Also, the STROBE checklist recommends not to use these terms in study design. This is either longitudinal descriptive study or a cohort study (as you tried to explore factors associated with treatment outcomes) using routinely collected secondary data. Kindly make necessary changes.

3. Under the setting sub-section, please provide the map of the Special Region of Yogyakarta. This will be helpful for international readers.

4. I am bit confused with your setting and study site. It is better to include everything under Study Setting. Maybe include Line 70 to 76 as General Setting and everything under 'study site' as Specific Setting.

5. Line 80 and 81, it is mentioned that project commenced in 2020, but the shorter regimen was available only in March 2022. Also, all the index patients from 2018 to 2022 were identified and included. This is bit confusing. Whether you started your field implementation only after shorter treatment regimen was available? When were the household contacts were listed, evaluated and initiated on treatment?

6. Line 86, who did this activity of contact listing and contact screening?

7. Line 92: How did you decide who gets TST and who gets IGRA? Also, only those with TBI were offered CXR or everyone underwent CXR?

8. Line 107: Can you please explain what exactly 'clinical review found that risk outweighed potential benefit' mean?

9. Line 117: What Grading system was used for ascertaining severity of adverse events.

10. Please clarify the confusion with timelines related to study population.

Results: The results also require major modification.

1. In the first paragraph, please provide, a) total number of index patients identified, b) total number of HHC listed, c) number of HHCs screened, d) number of HHC investigated for TBI, e) number wwho had contraindications, f) number diagnosed with active TB. You can look into the flow chart from this manuscript (https://www.mdpi.com/2414-6366/8/7/332). Without commenting on the above numbers, it is difficult to comment on the uptake.

2. Please describe the rate of adverse events (may in person-weeks), severity and management of events. This needs to be described before assessing the impact of AEs on the treatment outcomes   

3. Line 155. As it is a cohort study, please use risk ratios as measure of association and not the odds ratios. You can try binomial regression to get the risk ratios. 

Discussion: Discussion requires major revision.

1. Paragraph-1. I think you need to be very cautious in interpreting this uptake. Better not to over sell the results.

2. Please highlight the important limitations of the study before trying to discuss the implications of the study findings

3. Some of the things mentioned in the discussion are appearing for the first time in the manuscript (timing of AEs). Kindly highlight the findings in the results section before mentioning it in the discussion section.

4. There is need for discussing the study finding in context with programme setting. 

5. The whole manuscript misses on the duration from the treatment initiation of index patient and initiation of TPT in the HHCs

Reviewer 2 Report

Comments and Suggestions for Authors

This is an interesting study of real world experince of TB preventive treatment.

It is well presented with only a few minor comments needing clarified

Line 54 using shorter TPT regimens of daily isoniazid  you don’t say how long for

Line 56 under 5 should be under 5 years old or similar

Line 93 aged above 5 years

Line 94 under five year old with living with clinically confirmed index cases 

Line 99 to 102 repeat much of lines 92-5 please simplify

Line 146.  7 contacts were started 146 on 6H, 21 on 6LFX (6 months levofloxacin ± ethambutol), and 1 on 4R, all of them were 147 excluded from the study. 

I would re word as The remaining 29 were started on other regimes [7 contacts were started 146 on 6H, 21 on 6LFX (6 months levofloxacin ± ethambutol), and 1 on 4R} and were excluded from the study

Line 159 19 had not been evaluated  I am not clear what this means?  Were they lost to follow up or what. And in table 2.   This does not fit your outcomes in line 123

Line 160 1 failed the treatment This is not given as an outcome in line 125/6

Table 2 I don’t think you need the total column

Line 179 compare favorably to those observed and to results with longer regimen please give the values. Was this statistically significant or not?

Line 199 For most side effects, there was a slightly higher proportion of contacts on 3RH reporting the side effect than for contacts on 3HP Was this statistically significant or not?

Line 224 replication of its model of care is likely to be challenging if reliant on recurrent 224 health service funding. What requires the additional funding? Employment of the nurses who contact the contacts and what else?  
